# Energy Metabolism in Cancer: The Roles of STAT3 and STAT5 in the Regulation of Metabolism-Related Genes

**DOI:** 10.3390/cancers12010124

**Published:** 2020-01-03

**Authors:** Arturo Valle-Mendiola, Isabel Soto-Cruz

**Affiliations:** Molecular Oncology Laboratory, Cell Differentiation and Cancer Research Unit, FES Zaragoza, National University of Mexico, Batalla 5 de mayo s/n Colonia Ejército de Oriente CP, Mexico City 09230, Mexico; arturo.valle@unam.mx

**Keywords:** cancer metabolism, Warburg effect, transcription factors, HIF, STAT3, STAT5

## Abstract

A central characteristic of many types of cancer is altered energy metabolism processes such as enhanced glucose uptake and glycolysis and decreased oxidative metabolism. The regulation of energy metabolism is an elaborate process involving regulatory proteins such as HIF (pro-metastatic protein), which reduces oxidative metabolism, and some other proteins such as tumour suppressors that promote oxidative phosphorylation. In recent years, it has been demonstrated that signal transducer and activator of transcription (STAT) proteins play a pivotal role in metabolism regulation. STAT3 and STAT5 are essential regulators of cytokine- or growth factor-induced cell survival and proliferation, as well as the crosstalk between STAT signalling and oxidative metabolism. Several reports suggest that the constitutive activation of STAT proteins promotes glycolysis through the transcriptional activation of hypoxia-inducible factors and therefore, the alteration of mitochondrial activity. It seems that STAT proteins function as an integrative centre for different growth and survival signals for energy and respiratory metabolism. This review summarises the functions of STAT3 and STAT5 in the regulation of some metabolism-related genes and the importance of oxygen in the tumour microenvironment to regulate cell metabolism, particularly in the metabolic pathways that are involved in energy production in cancer cells.

## 1. Introduction

All cells need energy and for this purpose, they use macromolecules, which are degraded and thus, cells obtain the necessary energy for their essential functions. The central axis of energy metabolism consists of glycolysis, the Krebs cycle, and the respiratory chain. Glucose is the primary fuel and oxidises, via glycolysis, to provide energy for all cells (Figure 1). This metabolic pathway consists of 10 consecutive enzymatic reactions that convert glucose into two molecules of pyruvate, which connect with other metabolic pathways. During glycolysis, two net ATP and two net NADH molecules are produced. This pathway usually works with limited amounts of oxygen and is only an effective means of energy production during short, intense exercise, providing energy for a period of 10 s to 2 min (it is dominant for about 10–30 s during maximal effort). A by-product of glycolysis is lactic acid; this molecule accumulates in muscles and can produce tiredness and soreness. ATP is the primary energy source for performing metabolic work, while NADH can have different functions—it is the source of reducing power in anabolic reactions or, in the presence of oxygen, it can be oxidised in the respiratory chain.

The functions of glycolysis are in the generation of high energy molecules (ATP and NADH) as a source of cellular energy for aerobic respiration (in the presence of oxygen) and fermentation (without oxygen); the pyruvate generated in glycolysis enters the Krebs cycle and the intermediates of three and six carbons can be used in other metabolic processes. As shown in Figure 1, each reaction in glycolysis is catalysed by a specific enzyme. Many enzymes of the glycolytic pathway play significant roles in diverse non-glycolytic processes, such as transcriptional regulation, which enables cancer cells to meet other cellular demands. All enzymes that are involved in this pathway are deregulated in cancer [1,2,3,4,5,6,7,8,9,10,11,12,13,14,15,16,17,18,19,20,21,22,23,24,25,26,27,28,29,30,31,32,33,34,35,36,37,38,39,40,41,42,43,44,45,46,47,48,49,50,51,52].

A critical metabolic pathway that is parallel to glycolysis is the pentose phosphate pathway (also called the phosphogluconate pathway or hexose monophosphate shunt). It generates NADPH as well as ribose-5-phosphate; this molecule is a precursor for the synthesis of nucleotides. There are two distinct phases: the first is oxidative, in which NADPH is generated, and the second is the non-oxidative synthesis of 5-carbon sugars. This pathway is critical for cancer cells because it generates pentose phosphates, which support a high rate of nucleic acid synthesis. It also provides NADPH for both the synthesis of fatty acids and cell survival under stressful conditions with high levels of intracellular reactive oxygen species. There is increasing evidence that cancer cells modulate the flux of the hexose monophosphate shunt for their benefit since alteration of the pathway contributes directly to cell proliferation and survival. Like glycolysis, most enzymes of the pentose pathway are deregulated in cancer (Figure 2) [53,54,55,56,57,58,59,60,61,62].

The Krebs cycle, also known as the tricarboxylic acid cycle (TCA) or citric acid cycle (CAC), is fundamental in all cells that use oxygen during cellular respiration. Furthermore, it provides precursors (for example, ketoglutarate and oxaloacetate) for the production of amino acids as well as other fundamental molecules. The TCA cycle is composed of a series of enzymatic reactions occurring in the mitochondrial matrix and is a vital source of metabolic intermediates, providing energy, macromolecules, and redox balance to the cell. This is important for highly proliferating cells, like tumour cells, which require a continuous supply of small metabolites for the synthesis of proteins, nucleic acids, and lipids. Aberrant TCA cycle function is implicated in a wide variety of pathological processes such as obesity. Moreover, several TCA cycle enzymes are deregulated in cancer (Figure 3) [63,64,65,66,67,68,69,70,71,72,73,74,75,76,77,78,79,80,81,82].

The electron transport chain (ETC) controls combustion, which generates energy to be used to drive the oxidation of NADH, which produces three ATP molecules and FADH_2_ (which produces two ATP molecules) at the expense of reducing oxygen. The ETC is a series of sequential redox reactions in which the electrons are transported from one component to the next to reach the final acceptor, where the electrons reduce the oxygen to water. The phosphorylation of ADP to produce ATP is carried out by ATP synthase; this process is called oxidative phosphorylation and is the way that aerobic organisms obtain energy (Figure 4).

For a long time, these metabolic pathways were considered to be independent of the signal transduction pathways that regulate normal cell functions. However, increasing evidence exists to support the relationship between the signal transduction pathways induced by the growth factor, cytokines, and hormones, amongst others, and the regulation of energy metabolism. In cancer, these signalling pathways are deregulated to sustain altered proliferation, cell cycle regulation, and evade growth suppression, among others [83]. More recently, evidence points to the role of altered molecules to reprogram cellular metabolism [84]; for example, the role of transcription factors that regulate metabolic enzymes. Cytokines activate signal transducer and activator of transcription (STAT) proteins and mediate the metabolic switch in tumour transformation [85] (see Section 5).

## 2. Anaplerosis and Cataplerosis

The concept of anaplerosis describes a series of reactions or pathways that replace the pools of metabolic intermediates in the TCA cycle. The oxidation of acetyl coenzyme A to CO_2_ is the critical process in energy metabolism. Nonetheless, the TCA cycle also functions in anabolic pathways and its intermediates leave the cycle to be converted primarily to glucose, fatty acids, or amino acids (non-essential). After the removal of anions of the TCA cycle, they must be replaced to permit its continued function. This process is termed anaplerosis and the main anaplerotic enzyme is pyruvate carboxylase (this enzyme directly generates oxaloacetate in mitochondria) [86]. Cataplerosis balances anaplerosis by removing excess intermediates from the TCA cycle [87]; in fact, it is a set of opposite reactions to anaplerosis to increase the level of high-energy molecules (ATP) and thus, regulates the levels of TCA cycle intermediates. If intermediates can be added to the cycle, it is equally important to remove them to avoid the accumulation of anions in the mitochondrial matrix. There are several cataplerosis enzymes, including phosphoenol-pyruvate carboxykinase (PEPCK), aspartate aminotransferase, and glutamate dehydrogenase. The regulation of anaplerosis and cataplerosis depends on the metabolic and physiologic states and the specific tissue or organ involved. During starvation, cataplerosis, via phosphoenolpyruvate, supports gluconeogenesis, which may be regulatory in the liver, whereas in the kidneys, anaplerosis via the uptake of glutamine may be regulatory [86,88].

For example, in cancer cells, citrate is processed by ATP citrate lyase (ACL) to produce acetyl-CoA (this molecule can be used for fatty acid production). This phenomenon results in a truncated TCA cycle and produces a flow of metabolites out of the cycle, resulting in the augmentation of cataplerosis because the influx of metabolites needs to be balanced by anaplerosis (through the influx of metabolites). In many cancer cells, glutamine satisfies this function through conversion to glutamate and then to the intermediate α-ketoglutarate [89]. Cancer cells induce enzymes that process glutamine (glutaminase 1 and glutaminase C) and glutamate oxaloacetate transaminases to convert glutamate to α-ketoglutarate [90]. On the other hand, citrate can be moved to the cytosol, where it can be converted back to acetyl-CoA by ACL. Akt facilitates this diversion of mitochondrial citrate from the TCA cycle to acetyl-CoA production by phosphorylating and activating ACL [91,92,93]. Another metabolite of TCA, oxaloacetate, can be transaminated to produce aspartate, which can serve as a precursor for asparagine, and α-ketoglutarate can be transaminated to produce glutamate which, in turn, can be converted to proline, arginine, and glutamine. Most cancers depend on these syntheses rather than exogenous supplies [84]. Through α-ketoglutarate, glutaminolysis provides at least half of the NADH pool, and all FADH_2_ can potentially be generated via acetyl-CoA-independent reactions [94].

## 3. The Warburg Effect in Normal Cells

Healthy differentiated cells use the tricarboxylic acid cycle and oxidative phosphorylation to generate the energy and biomass that is necessary for normal cell function. The tricarboxylic acid cycle links the products of the oxidation of pyruvate and malate (produced in the cytosol) to CO_2_ with the generation of NADH to be further oxidised by the mitochondrial respiratory chain. This system transfers electrons to create a proton gradient across the inner mitochondrial membrane, which is used by the complex F_1_F_0_ ATPase to drive the synthesis of ATP. When cells present fundamental changes in nutrient metabolism and depend on aerobic glycolysis, this change is known as the Warburg effect [95,96,97].

Although the Warburg effect is frequent in tumour cells, some healthy cells also display this effect. The cells of the immune system are maintained in a quiescent state up until their activation. Good examples of enhanced glycolysis are M1 macrophages and T lymphocytes [98]. The M1 cells have increased glycolytic flux and reduced mitochondrial oxidative phosphorylation [99]. This metabolic change occurs in the context of an altered TCA cycle. M1 cells have specific TCA breakpoints at isocitrate dehydrogenase and succinate dehydrogenase, in which their gene expression is downregulated [100,101] similarly to tumour cells. During T lymphocyte activation, they go through several metabolic changes: rapid proliferation, synthesis of large amounts of a variety of effector proteins, and preparation to enter a potentially hypoxic environment. These metabolic adaptations resemble those observed in the cancer metabolism paradigm (proliferation, anabolism, energy production to generate building blocks, and the lack of nutrients and oxygen can limit the metabolic flux since they restrict metabolite access and oxygen) [102]. This phenomenon indicates that cancer cells only adopt metabolic strategies similar to those used by T lymphocytes in response to their activation; for this reason, cancer cells pervert these metabolic changes.

Glucose is introduced into the cell by glucose transporters and is metabolised to pyruvate in the cytosol by glycolysis; the net production is only two ATP molecules per glucose. In non-transformed quiescent cells, the pyruvate that is produced in the glycolysis is imported to the mitochondrial matrix, where it is converted to acetyl-CoA by the action of the pyruvate dehydrogenase complex (PDH). The acetyl-CoA enters the tricarboxylic acid cycle and the generated NADH is oxidised via oxidative phosphorylation. Compared to glycolysis, this process is highly efficient for the generation of ATP; the complete oxidation of one molecule of glucose produces 36 ATP molecules [103].

Some studies have demonstrated that the resting lymphocytes obtain most of their ATP by oxidative phosphorylation; nevertheless, within hours of stimulation, lymphocytes begin to increase glucose uptake up to 40-fold and secrete high amounts of lactate [104]. This increase in aerobic glycolysis precedes, and is essential for, the growth and proliferation of stimulated T cells [105,106].

Cancer cells do not increase glycolysis solely because their capacity for oxidative phosphorylation is saturated. Instead, aerobic glycolysis and basal oxidative phosphorylation provide sufficient energy to support the cell survival and growth demands of cancer cells and active T cells [102]. One advantage of glycolysis (and the TCA cycle) is the generation of molecular intermediates, which are used as carbon moieties to generate amino acids, lipids, and nucleotides; in this manner, a central function of aerobic glycolysis is to provide sufficient intermediates for biosynthesis in proliferation and cell growth. Glycolysis and the TCA cycle can supply both ATP and intermediates to multiple pathways to potentially support cells under stressful conditions. Furthermore, high rates of glycolysis can protect against apoptosis [107]. Moreover, glycolysis is independent of oxygen and adopting this type of metabolism can prepare cells to survive in a hypoxic environment. Adoption of a highly glycolytic metabolism may help both tumours and lymphocytes to survive and proliferate during conditions of low oxygen availability [102].

In the case of T cell activation, the induced metabolism is maintained by the continuous signalling of interleukin (IL)-2 (and other cytokines that share the γc chain, like IL-7). In fact, this cytokine can induce amino acid uptake and protein synthesis [108]. This effect is, in part, mediated directly by STAT5 [109]. JAK/STAT3 signalling in lymphocytes induces the expression of PIM kinases, which themselves participate in regulation of the AMP/ATP ratio and energy metabolism. Furthermore, the AMPK complex is capable of detecting changes in the concentrations of AMP and ADP; AMPK is also activated by glucose deprivation [95,110]. These effects lead to the modulation of the mTORC1 pathway and the control of cell growth [110]. The γc-receptor directs the maintenance of activated T cell metabolism and, therefore, potentially represents a useful tool to study the role of STAT-driven, PIM-mediated regulation of metabolism [95,102].

Despite significant metabolic similarities, the two systems—activated lymphocytes and cancer cells—have differences. For example, T cell metabolic reprogramming is transient and reversible. Unlike cancer cells with specific oncogenic mutations, the activated T cells are not malignant. In fact, following the clearance of infection, the majority of activated T cells will die due to the activation of apoptosis. Both activated T cells and tumour cells are kept alive by an imbalance of pro- and anti-apoptotic signals. This imbalance is maintained by cytokine signalling through Akt and other pathways and by glycolytic flux [96,97,102]. However, in cancer cells, this balance is maintained by glycolytic flux and oncogenic signalling, for example, hexokinase (HK) is the first enzyme in glycolysis and is a rate-limiting enzyme. There are four subtypes: Firstly, HKI-IV is overexpressed in tumour cells, enabling the cells to ensure that is energy is produced in hypoxia. The most commonly overexpressed subtype is HKII; this molecule has been observed in cervical cell lines (HeLa and SiHa) in normoxia and hypoxia [11], in hepatoma [111], breast [112], and brain [113] cancers. Thus, T cells provide a unique opportunity to understand how metabolism is used in healthy cells to achieve proliferation in comparison to that observed in cancer cells. Further research on the immune cell’s metabolism could provide the basis for new treatments targeting cancer metabolism.

## 4. Cancer Metabolism

Cancer cells frequently use well-established processes, like the Warburg effect, and adapt them for their benefit. Thus, some of the most remarkable characteristics of many types of cancer cells are the presence of altered metabolism, enhanced glucose uptake and glycolysis, and decreased oxidative metabolism. Tumour cells reprogram their metabolism to meet their high demand for energy; in fact, this change in metabolism is considered one of the hallmarks of oncogenic transformation [83]. This adjustment is essential for tumour cells to acquire sufficient energy to meet the anabolic demands necessary to generate the biosynthetic precursors that are required for cell growth and division.

In multicellular organisms, nutrient uptake is tightly regulated in the control of systems that prevent abnormal proliferation [83]. Nevertheless, tumour cells can exceed these metabolic restrictions, acquiring mutations in essential genes such as tumour suppressors and oncogenes. These genetic mutations may accumulate in the cells throughout the lifetime of an individual and change the function of signalling pathways that regulate metabolism. The unusual changes in these pathways increase nutrient uptake and alter metabolism to produce the necessary energy for survival and cell proliferation [114,115]. It has been reported that energetic metabolism, notably glucose metabolism, is connected to growth control by silencing of specific tumour genes, which drives uncontrolled cell proliferation, cycle arrest, and senescence [84,116,117,118]. Most of the proto-oncogenes and tumour suppressor genes code for molecules that participate in several signal transduction pathways, and their role in carcinogenesis has been related to their capacity for regulating the cell cycle to sustain proliferative signals that help cells to evade growth suppression and cell death [116]. Growing evidence exists that indicates that the main function of active oncogenes and inactive tumour suppressors is cell metabolism reprogramming [118]. In normal tissues, approximately 10% of cell energy is provided by glycolysis, while the aerobic respiratory chain that takes place in the mitochondria contributes 90% of cell energy. However, in cancer cells, approximately 50% of cell energy is generated by glycolysis and the rest of the energy is produced by the mitochondria [119]. This change is sustained even when the O_2_ is enough to maintain mitochondrial function (anaerobic glycolysis) [120]. Several reports suggest that there is an important correlation between the JAK/STAT pathway and cell metabolism—an aberrant pathway in cancer cells—since the constitutive activation of STAT proteins promotes glycolysis through the transcriptional activation of hypoxia inducible factors and, therefore, the alteration of the mitochondrial activity [121,122,123,124,125,126]. It seems that STAT proteins function as an integrative centre for different growth and survival signals for energy and respiratory metabolism and they have a central role in the metabolic function of the cell.

## 5. Role of the Transcription Factors STAT and HIF in the Deregulation of Energy Metabolism in Cancer Cells

### 5.1. Role of STAT Proteins in Metabolism

Signal transducer and activator of transcription (STAT) proteins are essential transcription factors for the cellular response to cytokines and growth factors [127,128,129]. Upon the binding of a ligand, the receptor becomes phosphorylated in tyrosine (pTyr); STAT proteins bind to the receptor and are phosphorylated by JAK kinases. They then dimerise, translocate into the nucleus, and regulate the expression of genes that modulate cellular functions (Figure 5). Increasing evidence suggests that STAT signalling may be involved in the regulation of cellular metabolism [130].

STAT molecules (particularly STAT3 and STAT5) are constitutively activated in a large variety of cancers. The actual model describes STAT molecules regulating the expression of nuclear target genes involved in proliferation, cell cycle progression, and resistance to apoptosis [130]. In Ras-mediated oncogenesis, STAT3 has been reported in mitochondria, and altered glycolytic and oxidative phosphorylation activities have been suggested, indicating that STAT proteins regulate an unknown metabolic function in mitochondria [131]. These results suggest that STATs might have supplementary functions in the mitochondria to alter cellular metabolism in ways that favour oncogenic transformation [130]. We discuss how the transcription factors STAT3 and STAT5 participate in the regulation of energy metabolism and their involvement in the regulation of HIF-1α, an important regulator in the cancer hypoxic microenvironment.

### 5.2. STAT3

The STAT3 transcription factor is well known for functioning as an anti-apoptotic factor, enhancing DNA repair in several malignancies, like leukemic stem cells [132], breast [133,134], prostate [132,133], fibrosarcoma, myelomas, lymphomas, head and neck, lung, pancreatic [134,135], and glioblastoma [136]. STAT3-dependent gene expression in cancer is heterogeneous, reflecting the implication of this factor in multiple steps of the oncogenic program [137]. In response to interferon β, pTyr-STAT3 participates in modulating mitochondrial electron transport chain (ETC) activity and oxidative phosphorylation [137]. Some observations suggest the existence of an essential correlation between STAT3 and cell metabolism; the latter process is aberrantly regulated in cancer cells. Evidence exists that constitutive activation of STAT3 promotes glycolysis through transcriptional induction of HIF-1α and decreased mitochondrial activity [85,128]. Activation of STAT3 precedes HIF-1 transcriptional response in oxygen deprivation; pSTAT3 has a peak after five minutes of oxygen deprivation, since maximum HIF-1α stabilisation requires 120 min [138]. In oesophageal squamous cell cancer, STAT3 bound to the HIF1α promoter and the knockdown of STAT3 inhibited epithelial–mesenchymal transition and downregulated HIF1α [139]. In malignant peripheral nerve tumours, STAT3 and HIF-1α promote oncogenic phenotypes, similarly to oesophageal cancer, where STAT3 knockdown was sufficient to block the expression of HIF-1α [140].

Exactly how STAT3 phosphorylation can be regulated within the mitochondria is not understood, however the phosphatase SHP2 was proposed as a potential player in dephosphorylating mitochondrial STAT3 [141]. Phosphorylation in serine 727 (S727) of STAT3 has emerged as a critical regulator of metabolic processes. STAT3 is localised in the mitochondria and some reports indicate that the molecule GRIM19, a component of Complex I of the ETC that was previously identified as a protein capable of interacting with STAT3, is necessary for the import of STAT3 into the mitochondria [142]. Inside the mitochondria, STAT3—phosphorylatedat S727—can interact with the Complexes I and II, which results in a reduction of reactive oxygen species (ROS) [130,143]. This function is essential under some stressful conditions, such as cardiac ischemia. In this case, the mitochondrial localisation of STAT3 protects the cells by preserving the activity of Complex I, reducing ROS production and the activation of caspase 3 [144]. STAT3 also interacts with cyclophilin D in the mitochondrial matrix, preventing the opening of the mitochondrial permeability transition pore (MPTP), avoiding apoptosis induced by calcium and necrosis [137]. STAT3 is capable of maintaining Ras-mediated carcinogenesis due to increased aerobic glycolysis and ETC activity, diminished opening of MPTP, and the lack of STAT3, which decreases the activity of Complex V (ATP synthase) dramatically [143].

Evidence exists that STAT3 affects the optimal activity of the ETC because it can modulate the energy status of the cell. For example, in pro-B cells and in murine hearts, mitochondria increase the activity of Complexes I and II [143]; in MEF cells, expressing the mutant H-RasV^12^, STAT3 increases the activity of Complexes II and V [131]; in MEF cells lacking SIRT1 (*SIRT1* KO), they show an increase in the activity of Complexes I, III, and IV and a decrease in Complex II activity [145]; in the murine cardiomyocyte cell line HL-1, this transcription factor elevates the activity of Complexes I–IV [146]. It seems that STAT3 functions as an integration centre for different survival and growth signals through its nuclear and mitochondrial activity [137]. Furthermore, the relative concentration of this molecule in the organelle is low; biochemical fractions reveal that 5–10% of total STAT3 localises in the mitochondria of multiple tissues and cell lines [143]. Nevertheless, how STAT3 carries out its activities within the mitochondria remains an open question.

### 5.3. STAT5

STAT-5 is considered to be an oncogene because it brings about the activation of cyclin D1, c-Myc, and Bcl-xl expression and is involved in promoting cell cycle progression, cellular transformation, and in preventing apoptosis [147]. Aberrant signalling of STAT3 and STAT5 is present in different solid tumours, like bladder, breast, colon, head and neck, cervical, and liver cancers, gliomas, melanomas, and haemopoietic malignancies [148,149,150,151]. Cholez et al. gave one of the first pieces of evidence linking STAT5 to oxidative metabolism (in pre-B cells). They used a proteomic approach to identify differential types of regulation in cells either expressing or not expressing a dominant negative form of STAT5A (NALM6Δ5). Among the 14 identified proteins, six were involved in the control of oxidative stress. The possible link between STAT5 and oxidative metabolism could be in the downregulation of transaldolase, glutathione synthetase, DJ-1, and thioredoxin domain-containing protein 9 (proteins 1–4) expression [152]. The downregulation of the levels of DJ-1 protein (known as Parkinson disease protein 7) in pre-B cells may, therefore, increase cell death susceptibility through oxidative metabolism and the upregulation of QPRTase (quinolatephosphoribosyltransferase) and DDAH2 (dimethylargininedimethylaminohydrolase). Some of these proteins control the levels of ROS [152]. Cancer cells could use this mechanism to modify their metabolism and adapt better to the tumour microenvironment. 

The JAK/STAT pathway (in particular, STAT5) could regulate glucose metabolism by driving the expression of Pyruvate dehydrogenase kinase (PDK). STAT5 mediates the expression of PDK4 in adipocytes in response to prolactin [153]. Chueh et al. demonstrated an interaction between STAT5 and a metabolic enzyme, PDC-E2, in the mitochondria. STAT5 exhibits unique DNA-binding activity. The presence of mitochondrial STAT5 in tumour cells and cytokine-stimulated cells also coincides with their metabolic shift towards aerobic glycolysis [130]. In a later report, the same group demonstrated that PDC-E2 might function as a coactivator in STAT5-dependent nuclear and mitochondrial gene expression. The model proposed by Chueh et al. for nuclear–mitochondrial crosstalk through cytokine-induced STAT5 and PDC-E2 interaction initiated by cytokine stimulation is that receptor dimerisation induces tyrosine phosphorylation of the associated JAK2 tyrosine kinase. Active JAK2 phosphorylates the cytoplasmic tails of receptor subunits. Cytoplasmic STAT5 is recruited to the receptor complex and phosphorylated in tyrosine by the JAK2 kinase. STAT5 proteins dimerise and translocate into the nucleus and the mitochondria. In the nucleus, tyrosine phosphorylated STAT5 binds to the promoter regions of distinct target genes. The associated PDC-E2 may work and collaborate with histone acetyltransferase (HAT) to enhance STAT5-dependent nuclear gene expression. On the other hand, the binding of tyrosine-phosphorylated STAT5 to the control region of mitochondrial DNA may modulate transcription initiated from the heavy strand promoter (HSP) and light strand promoter (LSP) [153,154].

The translocation of STAT5 into the mitochondria requires phosphorylation in specific tyrosine residues. The mitochondrial localisation of tyrosine-phosphorylated STAT5 was observed in leukemic T cells (these cells express a constitutively activated STAT5), and the presence of STAT5 increased in response to IL-2. By contrast, the cells that were not treated with IL-2 did not show the presence of STAT5 in the mitochondria. Once in the mitochondria, STAT5 interacts with the E2 protein, a component of the pyruvate dehydrogenase complex; it is also able to bind to the mitochondrial genome, precisely into the D-loop region [130]. The mitochondrial localisation of STAT5 suggests that it may be involved in mitochondrial gene regulation, also coinciding with the metabolic shift to aerobic glycolysis observed in T cells and leukaemias stimulated with cytokines. HIF-2α (a closely related HIF-1α isoform) is also a target gene of STAT5 in haemopoietic stem cells (HSC) [155]. The decrease in the expression of HIF-2α reduces the expansion of HSC induced by STAT5. Glucose uptake is enhanced in HSC cells expressing STAT5, and in these cells, HIF-2α is required for the upregulation of genes associated with glucose metabolism. In T cells, it has also been observed that STAT5 mediates glucose uptake [109]. Both isoforms of HIF (HIF-1α and HIF-2α) regulate the expression of numerous common genes, while HIF-1α preferentially induces genes of the glycolytic pathway [156,157]. HIF-2α is involved in the regulation of essential genes for tumour growth, cell cycle progression, and maintenance of the pluripotency of stem cells, like the proto-oncogene c-Myc41 and the stem cell factor OCT-3/4.

Chronic liver diseases and the development of hepatocellular carcinoma are tightly related and represent a real medical challenge as treatment options are minimal. Some studies using animals have shown that the genetic deletion of STAT5 in the liver is associated with a high susceptibility to fatty liver disease, fibrosis, and cancer, pointing to a protective role of hepatic STAT5 in mouse models of chronic liver disease [158]. Several studies have suggested that growth hormone (GH)-STAT5 signalling plays a vital role in controlling hepatic lipid metabolism. Mice with a liver-specific STAT5 ablation were shown to develop steatosis, glucose intolerance, insulin resistance, late-onset obesity, and impaired liver regeneration. Notably, the expression of genes associated with adipogenesis (PPARγ) and fatty acid uptake (CD36, or fatty acid translocase) were upregulated in Stat5a/b-deficient mice. These changes may partially explain steatosis induced by loss of STAT5 [159,160]. Barclay et al. demonstrated that deficient GH-dependent STAT5 signalling correlates with steatosis through microarray analysis, quantitative PCR, and chromatin immunoprecipitation identified putative targets of STAT5 (FA (fatty acid) synthetase, CD36 signalling) that are responsible for the steatosis associated with a healthy diet [159]. The liver is a target organ for steroid hormone metabolism. Several studies have suggested that GH-STAT5 regulates genes linked to steroid metabolism. One of these genes, *HSD3b5*, catalyses the formation of the poorly active metabolite androstanediol from dihydrotestosterone [160]. *HSD3b5* gene expression was shown to be downregulated in the livers of STAT5-deleted male mice [161,162]. By contrast, an alternative gene involved in testosterone metabolism, testosterone 16α-hydroxylase (Cyp2b9), which hydroxylates testosterone at the 16α position, was found to be upregulated in STAT5-deleted male mice. Cyp7b1 (oxysterol 7α-hydroxylase), which is responsible for the hydroxylation of dehydroepiandrosterone, androstenediol, and 25-hydroxycholesterol, was found to be downregulated in STAT5-deleted male mice [163].

Another critical organ in which the STATs play an essential role is the heart; JAK1, JAK2, TYK2, and all the members of the STAT family are expressed in the heart [164]. Multiple studies have demonstrated a favourable and protective role of STAT3 in the heart. This role has mainly been pointed out using data from animal experiments. As STAT3 knockout mice were shown to have early embryonic lethality [165], specific cardiac myocyte *STAT3* knockout mice have been a helpful tool to investigate the role of STAT3 in the heart. The use of these mice and the pharmacological inhibitor of JAK2 (AG490) demonstrated a protective and anti-apoptotic role of STAT3. This role has mainly been demonstrated using a model of ischemia/reperfusion injury [166]. Similar cardio-protective effects have been described for STAT5 activation [167]. Heusch et al. used a model of cardio-protection by remote ischemic preconditioning (RIPC), and only STAT5 activation was associated with protection by RIPC. The available data suggest that the functions of STAT3 and STAT5 might have different roles in pigs and humans; STAT5, but not STAT3 activation, is associated with protection in human hearts, whereas STAT3 activation and possibly STAT5 inhibition are associated with protection in pigs. Alternatively, different procedures and protocols of cardio-protection may recruit a different pattern of STAT isoform signalling [167]. Tumour cells could use this mechanism to evade apoptosis and increase their resistance to stress caused by a low oxygen concentration. STAT5 undergoes nuclear and mitochondrial activities, and both functions cooperate to induce metabolic shift, however the role of this transcription factor in the mitochondria is unknown and remains an open question. 

As mentioned above, the STATs are essential for promoting glycolysis through the activation of HIF molecules. For example, STAT3 induces the expression of HIF-1α, augments glycolysis, and decreases mitochondrial activity. Furthermore, STAT5 can induce the expression of HIF-2α, which is necessary for the upregulation of genes associated with glucose metabolism. Thus, it is clear that STAT proteins are involved in regulating hypoxia-inducible factors that are the main regulators of oxygen homeostasis. Hypoxia is a critical microenvironmental factor that is related to progression and metastasis. Most of the solid tumours show hypoxic conditions, and this minor concentration of O_2_ induces an altered regulation in gene transcription, leading to changes in cell metabolism [168,169].

### 5.4. The Importance of Oxygen and HIF-1α in the Regulation of Metabolism

Oxygen is a basic necessity for life as it is the final acceptor of electrons in the ETC; its relative abundance or absence can modify processes like embryogenesis, wound healing, and stem cell maintenance [170,171,172,173]. In fact, because of its physiological importance, oxygen levels and molecular pathways regulated by oxygen are strictly regulated, so failure in the proper regulation in response to low oxygen (hypoxia—an oxygen concentration of approximately 6% or lower generates hypoxic stress) can promote a number of diseases, including diabetic retinopathy, ischemic heart disease, and cancer [170,174,175]. In cancer, the microenvironment of tumours is related to chronic hypoxia [175,176]. In response to low levels of O_2_, the tumour cells turn on specific genes that promote adaptation to hypoxic stress, including those involved in angiogenesis, cell survival, proliferation, evasion of growth suppressors, metastasis, metabolic reprogramming, and mortality [177,178]. It is well documented that hypoxia promotes tumorigenesis and contributes to a poor clinical prognosis [179,180].

Hypoxia inducible factor-1alpha (HIF-1*α*) is a critical molecule that is overexpressed in many types of cancer; this transcription factor forms heterodimers consisting of an oxygen-dependent subunit (subunit α) and an independent oxygen subunit (subunit β). There are three isoforms of the α subunit (HIF-1α, HIF-2α, and HIF-3α) and three isoforms of the β subunit, also known as the aryl-hydrocarbon receptor nuclear translocators (Arnt1, Arnt2, and Arnt3). Subunit α translocates into the nucleus and under normal conditions, has an extremely short half-life—less than five minutes [123]. The degradation of this subunit is mediated by the oxygen-dependent degradation domain (ODDD) positioned within the N-TAD, which contains specific proline residues (Pro402 and Pro564 in HIF-1α and Pro405 and Pro531 in HIF-2α) that are hydroxylated in an average oxygen concentration by a particular group of prolyl hydroxylases (PHDs) [124]. However, the α-subunit can dimerise with the β-subunit (this subunit is insensitive to oxygen and hence, is constitutively expressed) when the oxygen concentration is under 6% [124,125]; this complex binds to consensus sequence 5′-RCGTG-3′, which is present in the enhancer of the hypoxia response element (Figure 6).

High levels of expression of HIF-1α are associated with high mortality in cancer [181]; HIF is capable of inducing transcription of the pyruvate dehydrogenase kinases (PDK) 1 and 3 [182,183]. The PDK phosphorylates and inactivates the pyruvate dehydrogenase (PDH), preventing pyruvate from entering into the Krebs cycle, hence reducing mitochondrial oxygen consumption and decreasing the production of reactive oxygen species (ROS). HIF promotes the conversion of pyruvate to lactate through lactate dehydrogenase. This enzyme is a tetramer that includes the H subunit (LDH-H; this subunit is expressed ubiquitously by the gene IDHB). The LDH-H that is found in the heart is more active at low concentrations of pyruvate and is strongly inhibited by an excess of pyruvate (approximately 10^−2^ M). The subunit M (LDH-M, expressed by the gen IDHA) maintains its activity at relatively high pyruvate concentrations [184]. The IDHA gene is a direct target of HIF and is highly inducible by hypoxia. HIF, therefore, promotes the formation of a complex formed exclusively by LDH-M, which is more efficient in converting pyruvate to lactate; this results in a decreased flow of pyruvate into mitochondria [182].

HIF-1α is expressed in cervical cancer, which represents an early event in tumour development and is expressed in both normoxia and hypoxia conditions. The levels of glucose transport protein 1 (GLUT1) gradually increases during the transition from normal cervix conditions to cervical intraepithelial neoplasia; the expression of this molecule is related to lymph node metastasis, and the expression levels of GLUT1 and HIF-1α are correlated, indicating that HIF-1α may regulate the expression of downstream genes that are involved in the energy supply [114]. On the other hand, the contribution of HIF-1α to the regulation of hypoxic cell survival or death remains controversial. It has been reported that this molecule can have either pro-apoptotic or anti-apoptotic effects in different systems [185]. During severe hypoxia (or anoxia), apoptotic cell death is crucial to avoid hypoxia-induced mutations in cells. One proof of hypoxia-induced apoptosis is the suppression of the electron transport chain on the inner membrane of mitochondria [186]. Reports exist that HIF-1α can initiate hypoxia-mediated apoptosis by enhancing the expression of several genes (Bcl-2, p53, and others) [187]. However, during hypoxia, the translocation of the Bax protein to the mitochondria is downregulated and the inhibitor of apoptosis protein 2 (IAP-2) levels are upregulated; this combination of effects preserves the mitochondrial integrity and may help with cell survival during hypoxia. HIF-2 also plays an essential role during hypoxia by positively regulating cell proliferation, which is controlled by Myc [121,186]. In fact, during severe or prolonged hypoxia, the vast majority of the cells undergo apoptosis; nevertheless, some of the cells adjust to the environment and survive by avoiding necrosis and apoptosis, therefore resulting in an aggressive phenotype. This phenomenon suggests that cells that are non-sensitive to apoptosis in a tumour will be resistant to anticancer treatments [188,189].

## 6. Conclusions

The progress made in recent years in cancer research metabolism has improved our understanding of how aerobic glycolysis and other metabolic abnormalities that are observed in cancer cells support the anabolic requirements associated with cell growth and proliferation. There is increasing evidence that anabolism is under complex control, which is regulated directly through signalling induced by growth factors; all promote changes in aerobic glycolysis that are characteristic of the Warburg effect. Aberrant activation of the JAK/STAT pathway has been found in several hematologic malignancies and solid tumours. In particular, STAT proteins participate in cellular respiration and increasing evidence indicates that activated STATs can regulate energy metabolism by influencing the expression of critical enzymes that are important in these metabolic pathways. In this review, we discussed the role of STAT proteins in the regulation of metabolic enzymes. We focused mainly on cytoplasmic/nuclear STAT proteins, the fact that STAT3 and STAT5 are found in mitochondria, and that the effects on the regulation of metabolic enzymes are mainly mediated by an increased HIF-1α expression. The impact of STAT proteins on energy metabolism remains an open question, highlighting this as an important area for future research.

Despite the progress that has been made in recent years, it has been difficult to change the general idea that alterations in metabolism are an indirect phenomenon in cancer, a small side effect that pales in importance compared with the activation of the first signals of proliferation and survival. Therefore, altered metabolism should be considered a central aspect in the development and growth of tumours to which more attention must be devoted, since there is still much research to be done in this area of cancer. This phenomenon indicates that the cancer cells only adopt metabolic strategies similar to those used by T lymphocytes in response to their activation; for this reason, the cancer cells pervert these metabolic changes.

## Figures and Tables

**Figure 1 cancers-12-00124-f001:**
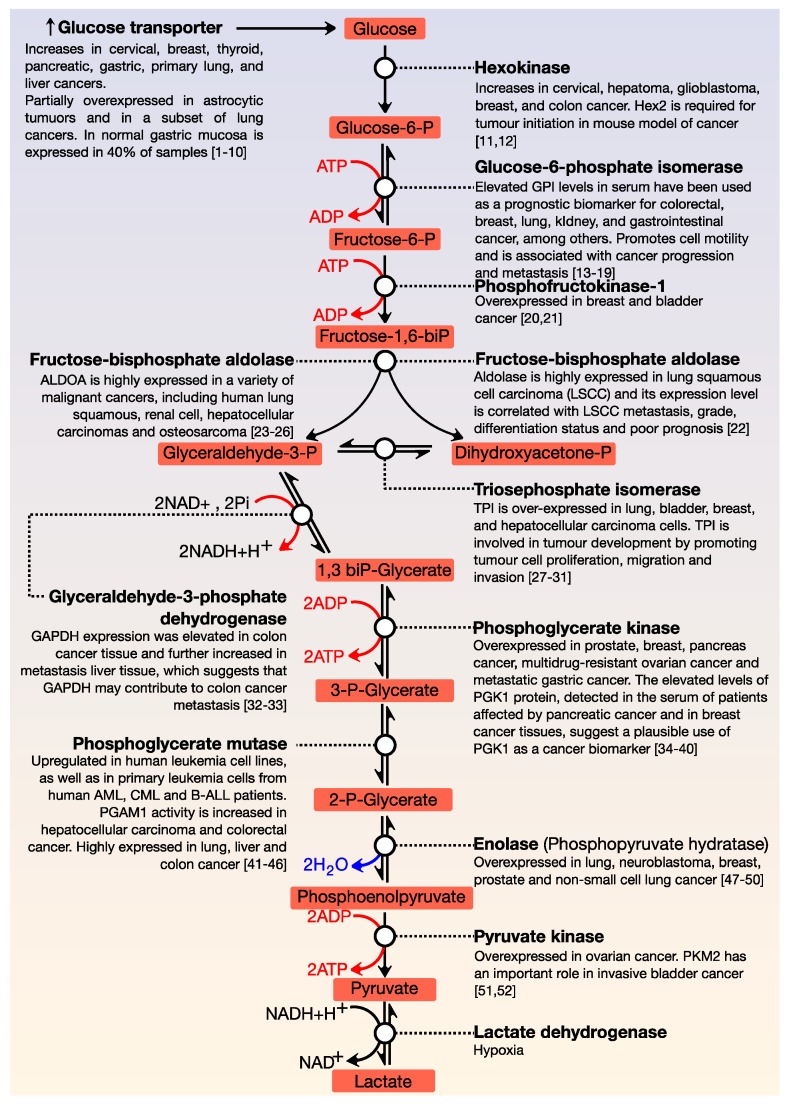
General view of glycolysis. The main steps of the regulation of this pathway are the conversion of glucose to glucose-6-phosphate; fructose-6-phosphate to fructose 1, 6-biphosphate; and the formation of pyruvate from phosphoenolpyruvate. All glycolytic enzymes are deregulated in cancer.

**Figure 2 cancers-12-00124-f002:**
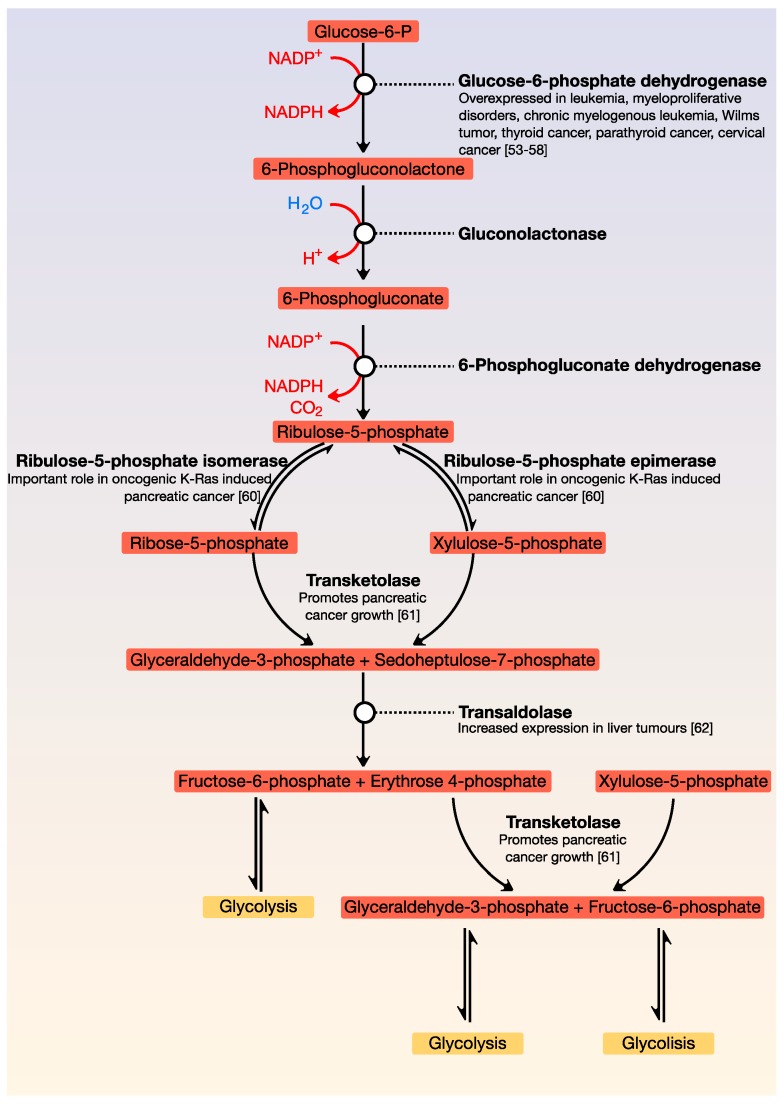
General view of the pentose phosphate pathway. This pathway is parallel to glycolysis and is the main source of NADPH and pentoses. Its point of regulation is the conversion of glucose-6-phosphate to 6-phosphogluconolactone.

**Figure 3 cancers-12-00124-f003:**
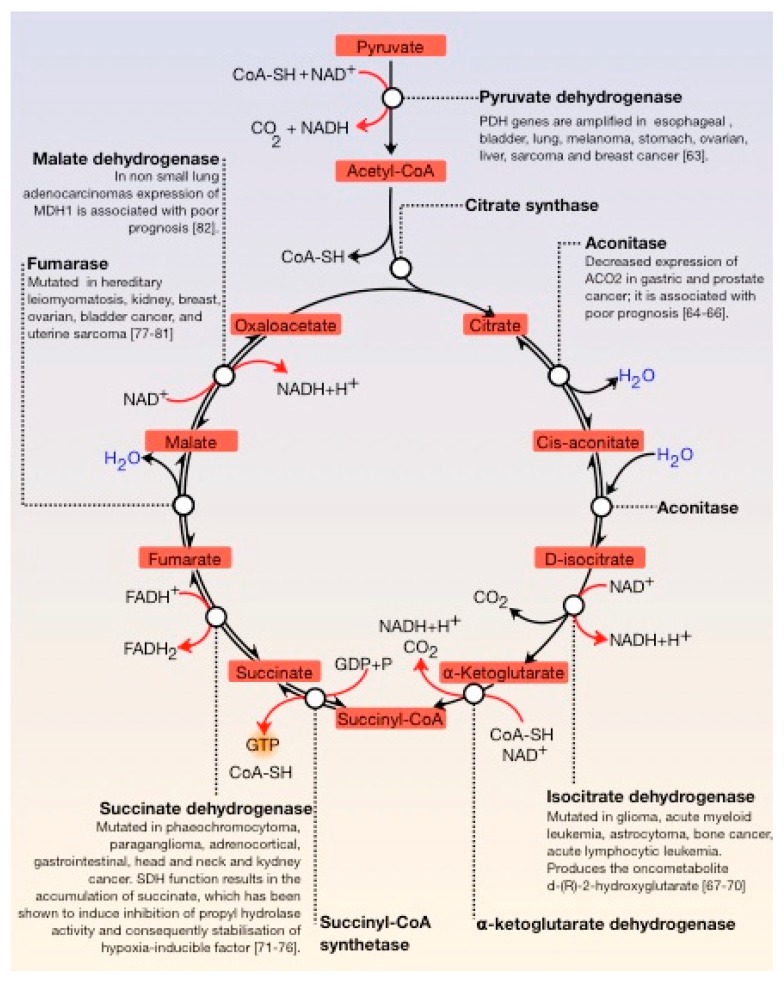
General view of the Krebs cycle. The regulation points are the condensation of oxaloacetate with acetyl-CoA, the conversion of D-isocitrate to α-ketoglutarate, and the conversion of this molecule to succinyl-CoA. Several enzymes of the tricarboxylic acid (TCA) cycle are deregulated in cancer. Several enzymes of the tricarboxylic acid (TCA) cycle are deregulated in cancer.

**Figure 4 cancers-12-00124-f004:**
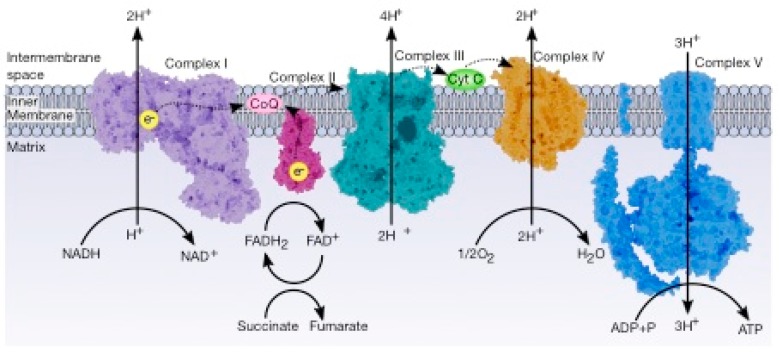
General view of the electron transport chain. This pathway is a series of five complexes transferring electrons from donors (NADH or FADH_2_) to acceptors (final acceptor is O_2_) and generates an electrochemical gradient which drives the synthesis of ATP.

**Figure 5 cancers-12-00124-f005:**
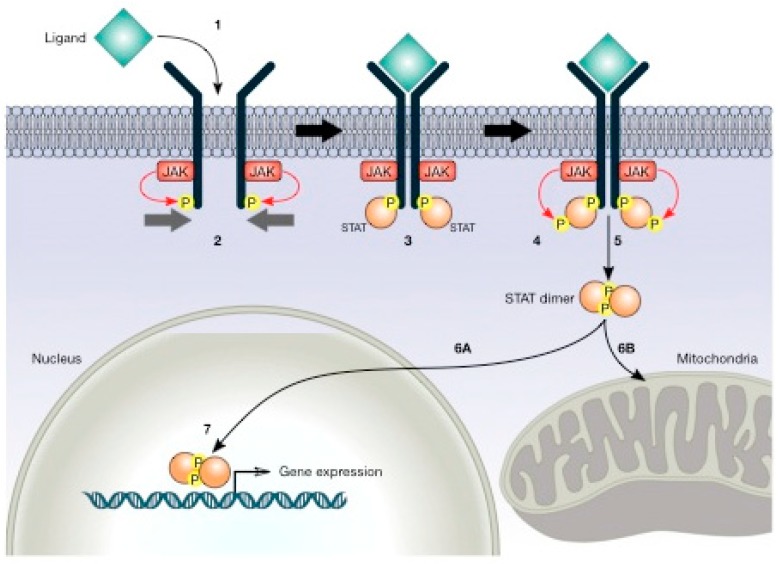
JAK/STAT (signal transducer and activator of transcription) pathway. This pathway initiates with ligand binding (1), which induces receptor dimerisation, and the associated JAKs phosphorylate the receptors (2). STATs bind to the phosphorylated receptor (3). Subsequently, JAKs phosphorylate STATs (4). STATs separate from the receptor and dimerise (5) and then migrate to the nucleus (6A) or mitochondria (6B). STATs regulate gene expression in the nucleus (7).

**Figure 6 cancers-12-00124-f006:**
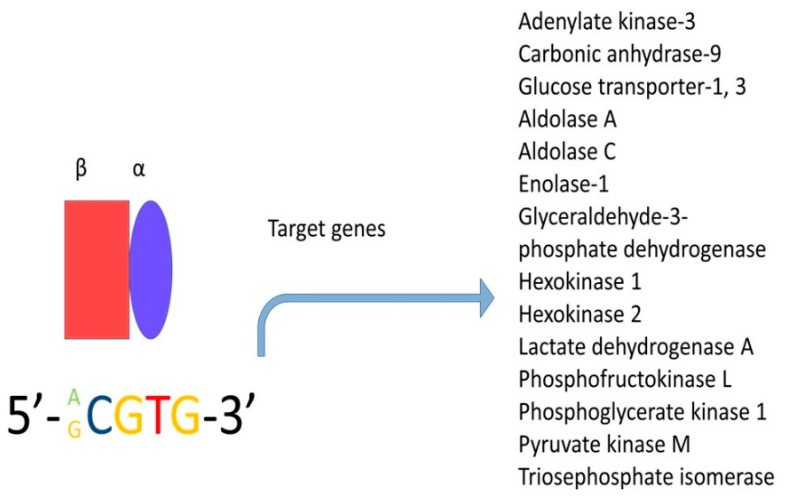
Hypoxia-inducible factor (HIF) is a transcription factor and is expressed in all metazoan organisms. This factor is composed of two subunits: HIF-1α and HIF-1β. Under hypoxic conditions, HIF regulates the expression of hundreds of genes. In this figure, only the genes related to metabolism are shown.

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
