# Peer review of "Energy Metabolism in Cancer: The Roles of STAT3 and STAT5 in the Regulation of Metabolism-Related Genes"

_cancers, 2020, doi:10.3390/cancers12010124_

Round 1

Reviewer 1 Report

This manuscript demonstrates a review of energy metabolism in cancer and the role of transcription factors in the regulation of metabolism-related genes. In my opinion, major revision is needed before publication. The following comments might be useful for resubmission.

1. The structure of the manuscript has not been made in a style of review article. Manuscript is not well organized, and it needs to be more polished.

2. Some languages and words are not suitable in this context, eg; page 4, line 84: “After the muscle returns to exercise, within two minutes of exercise”, line 87: “contraction in explosive and more extended time response” etc. The English language should be more constructive.

3. Abstract part should be more informed about this review article. In the abstract, authors have mentioned about p53, but there is no detailed discussion about this in the rest of the manuscript.

4. The pivotal role of the STAT protein in altered cancer cell metabolism should be discussed in a more detailed manner.

5. The link between metabolism and transcription factors should have been established in a better matter in the introduction part.

6. Pictorial representations are very poor. Colors and style should be more professional and authentic. In figure 2, enzyme name is missing for the conversion from 6-Phosphogluconate to Ribulose-5-phosphate. Some of the conversions are reversible and they should have a reversible arrow. In figure 3, the Krebs cycle should be presented in a circular structure rather than a random one. The fate of Succinate should be mentioned as it is important in further metabolism.

7. How anaplerosis and cataplerosis are relevant in this context? Just a description is not enough.

8. During discussing the Warburg effect, the hypothesis should be defined in short at the very beginning of this portion.

9. The full form of HIF-1a should be mentioned before using the abbreviated one. What is hypoxia, hypoxic stress and how they are related in metabolism? More comments will be useful in the role of HIF-1a in cellular adaptation to hypoxic stress.

10. Some links are missing between different parts of this manuscript, eg; no connection is found between parts 4 and 5 in page 7.

11. Short description of H and M subunits of LDH should be included where they have been mentioned. How apoptosis is important in altered cancer cell metabolism?

12. More references should be included which are relevant to this article.

Author Response

Reviewer 1

Comments and Suggestions for Authors

This manuscript demonstrates a review of energy metabolism in cancer and the role of transcription factors in the regulation of metabolism-related genes. In my opinion, major revision is needed before publication. The following comments might be useful for resubmission.

The structure of the manuscript has not been made in a style of review article. Manuscript is not well organized, and it needs to be more polished.

Response: To attend the reviewer’s suggestion, we have re-organized the article, adding information to make clear relationships between transcription factors and cancer biology.

Some languages and words are not suitable in this context, eg; page 4, line 84: “After the muscle returns to exercise, within two minutes of exercise”, line 87: “contraction in explosive and more extended time response” etc. The English language should be more constructive.

Response: We do agree with the reviewer. This paragraph was not clear and was not relevant to the description of the metabolic pathway; therefore, we eliminated it.

Abstract part should be more informed about this review article. In the abstract, authors have mentioned about p53, but there is no detailed discussion about this in the rest of the manuscript.

Response: We agree with the reviewer’s comment, and for this purpose, we added information to the abstract for clarity’s sake.

The pivotal role of the STAT protein in altered cancer cell metabolism should be discussed in a more detailed manner.

Response: To attend the reviewer’s suggestion, we discussed in a more detailed manner the role of the STAT proteins in the metabolism of cancer cells.

The link between metabolism and transcription factors should have been established in a better matter in the introduction part.

Response: To attend the reviewer’s suggestion, we linked metabolism and transcription factors in the cancer biology section of the article.

Pictorial representations are very poor. Colors and style should be more professional and authentic. In figure 2, enzyme name is missing for the conversion from 6-Phosphogluconate to Ribulose-5-phosphate. Some of the conversions are reversible and they should have a reversible arrow. In figure 3, the Krebs cycle should be presented in a circular structure rather than a random one. The fate of Succinate should be mentioned as it is important in further metabolism.

Response: We agree with the reviewer’s comment; thus, the pictorial representations are more professional and authentic, including the missing information. We have mentioned the importance of Succinate’s fate in cell metabolism.

How anaplerosis and cataplerosis are relevant in this context? Just a description is not enough.

Response: To attend the reviewer’s suggestion, we included information to stress the importance of these processes.

During discussing the Warburg effect, the hypothesis should be defined in short at the very beginning of this portion.

Response: To attend the reviewer’s suggestion, we defined the Warburg effect briefly at the beginning of section 3.

The full form of HIF-1a should be mentioned before using the abbreviated one. What is hypoxia, hypoxic stress and how they are related in metabolism? More comments will be useful in the role of HIF-1a in cellular adaptation to hypoxic stress.

Response: To attend the reviewer’s suggestion, we mentioned the full form of HIF factors before using the abbreviated one. We have defined hypoxia and the importance of oxygen in cancer metabolism. Also, we have mentioned the role of HIF-1a in cellular adaptation to hypoxic stress in cancer cells.

Some links are missing between different parts of this manuscript, eg; no connection is found between parts 4 and 5 in page 7.

Response: To attend the reviewer’s suggestion, we have included the connection between section 4 and section 5.

Short description of H and M subunits of LDH should be included where they have been mentioned. How apoptosis is important in altered cancer cell metabolism?

Response: To attend the reviewer’s suggestion, we have included a short description of the H and M subunits of LDH. Also, we have stressed the importance of apoptosis in cancer cell metabolism.

More references should be included which are relevant to this article.

Response: We have attended the reviewer’s suggestion and included some references relevant to the article.

Reviewer 2 Report

The manuscript by Valle-Mendiola and Soto-Cruz reviews some of the major metabolic pathways that are deregulated in cancer and addresses the salient transcriptional pathways that mediate these changes.

Major comments

Although logically organized, the level of English is extremely poor and distracting and must be improved enormously in order for the manuscript to be acceptable for publication.This reviewer would recommend that the authors engage the assistance of a native English speaker to edit the entire paper

2. The authors should mention that the “top” and “bottom “ portions of the TCA Cycle need not be coordinated and that the latter can be dissociated from the former. Indeed, during the anaplerotic conversion of glutamine to a-ketoglutarate, it is not necessary for the glycolytic pathway to be providing AcCoA. Thus, TCA Cycle reducing equivalents (NADH and FADH2) can be generated in an AcCoA-independent manner allowing Oxphos to continue under conditions where the supply of AcCoA derived from glucose or fatty acid oxidation might otherwise be compromised  (Wang et al. J Biol Chem. 2019 Apr 5;294(14):5466-5486.) During the discussion of how the JAK/STAT pathway upregulates T-cell metabolism via the mToR pathway (lines 153-156), the authors should mention that the pathways by which occurs mediated by AMPK; they should cite at least one the numerous excellent reviews on this topic, particularly those of Hardie et al (ex. Lin and Hardie. Cell Metab. 2018 Feb 6;27(2):299-313.).

Reviewer 3 Report

This review article is on an important topic: transcriptional regulation of cancer cell metabolism. However, several areas need to be improved:

1. While the title is about "role of transcriptional factors in...", the main focus is really just on HIF and STAT3/5. Therefore the title needs to be more specific instead of using the general term of "transcriptional factors".

2. It is unclear why HIF and STATs are particularly the two main subjects of this article. There is no particular connection between these two sets of transcriptional factors.

3. Major re-organization is needed. The arrangement of the article is rather unbalanced. Too much time is spent on the description of the metabolic programs (glycolysis, OxPhos, PPP, etc). The description on the transcriptional factors is rather short and brief, and again it is not clear why HIF and STATs are particularly the main focus. Significant amount of work is needed to reduce the details of the metabolic programs, to increase the description of the transcriptional factors, and to better integrate them into the context of cancer biology.

4. English needs improvement. There are many grammatical errors and improper usage of words and phrases. Paragraphs can also be re-organized to make the article more concise.

Author Response

Reviewer 3

Comments and Suggestions for Authors

This review article is on an important topic: transcriptional regulation of cancer cell metabolism. However, several areas need to be improved:

While the title is about "role of transcriptional factors in...", the main focus is really just on HIF and STAT3/5. Therefore the title needs to be more specific instead of using the general term of "transcriptional factors".

Response: To attend the reviewer’s suggestion, we changed the title to be more specific, including STAT3 and STAT5 in the title.

It is unclear why HIF and STATs are particularly the two main subjects of this article. There is no particular connection between these two sets of transcriptional factors.

Response: To attend the reviewer’s suggestion, we have described the connection between the STAT proteins and HIF, since HIF is under the regulation of STATs in the hypoxic tumour microenvironment.

Major re-organization is needed. The arrangement of the article is rather unbalanced. Too much time is spent on the description of the metabolic programs (glycolysis, OxPhos, PPP, etc). The description on the transcriptional factors is rather short and brief, and again it is not clear why HIF and STATs are particularly the main focus. Significant amount of work is needed to reduce the details of the metabolic programs, to increase the description of the transcriptional factors, and to better integrate them into the context of cancer biology.

Response: To attend the reviewer’s suggestion, we re-arranged the article. We reduced the details of the metabolic pathways, but we are not eliminating all the information since we are stressing the participation of the enzymes in tumour biology. Also, we increased the description of the transcription factors, integrating them into the context of cancer biology.

English needs improvement. There are many grammatical errors and improper usage of words and phrases. Paragraphs can also be re-organized to make the article more concise.

Response: To attend the reviewer’s suggestion, we had the article edited by a native English speaker.

Also, we re-organized some paragraphs to make the article more concise.

Round 2

Reviewer 1 Report

The authors have addressed the concern outlined in my previous review.

Reviewer 3 Report

The manuscript is improved, but still in a poor condition that does not justify publication. The manuscript is extremely difficult to follow with the poor language usage, which also makes it difficult for me to carefully assess the scientific accuracy. I list only some defects below, which are far from being thorough.

Despite the editing by a “native English speaker”, the language usage (both linguistically and scientifically) is still poor. There are many misuses of “the”, “and”, as well as many other grammatical errors. Many sentences are incorrectly constructed in the scientific context. It is highly suggested that the manuscript should be edited by a scientist in the filed with good language skill. The comma after “STAT3” in the title should be removed. The organization of the paper is still poor. Is the comparison between cancer cells and T cells a main point of the paper? The connection between HIF and STATs is still not clear and not well justified. The authors start talking about STATs before introducing them. What is the main point of the “Conclusion”? How come STATs and HIF are not even mentioned in the conclusion?

Author Response

Comments and Suggestions for Authors

The manuscript is improved, but still in a poor condition that does not justify publication. The manuscript is extremely difficult to follow with the poor language usage, which also makes it difficult for me to carefully assess the scientific accuracy. I list only some defects below, which are far from being thorough.

Despite the editing by a “native English speaker”, the language usage (both linguistically and scientifically) is still poor. There are many misuses of “the”, “and”, as well as many other grammatical errors. Many sentences are incorrectly constructed in the scientific context. It is highly suggested that the manuscript should be edited by a scientist in the filed with good language skill. The comma after “STAT3” in the title should be removed. The organization of the paper is still poor. Is the comparison between cancer cells and T cells a main point of the paper? The connection between HIF and STATs is still not clear and not well justified. The authors start talking about STATs before introducing them. What is the main point of the “Conclusion”? How come STATs and HIF are not even mentioned in the conclusion?

Response: To attend the reviewer’s suggestion, we had the manuscript edited by a specialist from MDPI editing services. We include the editing certificate.

Research in cancer metabolism has increased our understanding of the metabolic requirements of proliferating cells, as well as the metabolic alterations that promote tumour growth. It is becoming clear that signalling pathways activated by surface receptors can influence energy metabolism as part of their program of action. One of these signalling pathways, the JAK/STAT pathway is active in immune cells and is aberrant in cancer cells, and several reports exist that describe the relationship of this aberrant pathway and energy metabolism. Therefore, in this paper, we discuss the roles of STAT3 and STAT5 in the regulation of metabolism-related genes.

We consider that the paper is logically organized; it gives an introduction to the primary energy pathways pointing the enzymes that dysregulate in cancer; then, we describe the Warburg effect in healthy cells. In this section, we describe some studies that examine the metabolism of immune cells, mainly T cells. Several reports show that metabolic pathways connect to cell surface receptors of the immune system via signal transduction pathways; all promote changes in aerobic glycolysis characteristic of the Warburg effect. Activation and differentiation into effector cytotoxic T lymphocytes require a switch to glycolytic metabolism. The metabolic phenotype of active T lymphocytes is remarkably similar to cancer cells and serves the same purpose: to support macromolecule synthesis, rapid proliferation and high ATP demand. Thus, T cells provide a unique opportunity to understand how metabolism is used in healthy cells to achieve proliferation in comparison to that observed in cancer cells. Further research and a more comprehensive understanding of how metabolism regulates immune cell fate could provide the basis for new treatments targeting cancer metabolism.

We then discuss cancer metabolism, highlighting that energy metabolism connects to growth control by silencing of specific tumour genes which drive uncontrolled cell proliferation, cycle arrest, senescence, and cell metabolism reprogramming. Several reports suggest that there is an important correlation between the JAK/STAT pathway and cell metabolism since the constitutive activation of STAT proteins promotes glycolysis through the transcriptional activation of hypoxia-inducible factors and, therefore, the alteration of the mitochondrial activity. It seems that STAT proteins function as an integrative centre for different growth and survival signals for energy and respiration metabolism, and they have a central role in the metabolic function of the cell.

The final section includes an overview of the JAK/STAT signalling to introduce the importance of STAT3 and STAT5 in the regulation of metabolism-related genes, in particular, the HIF proteins, which have a central role in the upregulation of genes associated with glucose metabolism in the presence of low oxygen conditions. Hypoxia is a critical microenvironmental factor related to progression and metastasis. Most of the solid tumours show hypoxic conditions leading to changes in cell metabolism.

Finally, we conclude that metabolism is regulated directly through signalling induced by growth factors. Increasing evidence exists to support the relationship between the signal transduction pathways induced by the growth factor, cytokines, and hormones, amongst others, and the regulation of energy metabolism. In cancer, these signalling pathways are deregulated to sustain altered proliferation, cell cycle regulation, evade growth suppression, and cell metabolism reprogramming. More recently, evidence points to the role of altered molecules to reprogram cellular metabolism; for example, the role of transcription factors that regulate metabolic enzymes. Cytokines activate signal transducer and activator of transcription (STAT) proteins and mediate the metabolic switch in tumour transformation. The impact of STAT proteins on energy metabolism remains an open question, highlighting this as an important area for future research.